# Bond Properties of Steel Bar in Concrete under Water Environment

**DOI:** 10.3390/ma12213517

**Published:** 2019-10-26

**Authors:** Li Song, Fulai Qu, Guirong Liu, Shunbo Zhao

**Affiliations:** 1Yellow River Institute of Hydraulic Research, Zhengzhou 450003, China; songli@yeah.net; 2School of Civil Engineering and Communication, North China University of Water Resources and Electric Power, Zhengzhou 450045, China; liugr@ncwu.edu.cn (G.L.); sbzhao@ncwu.edu.cn (S.Z.)

**Keywords:** bond property, deformed steel bar, water environment, concrete, bond–slip curve, constitutive model

## Abstract

The present study concerns the bond behavior of steel bar in concrete under a water environment. This topic was put forward because of the changes of concrete under a water environment and the importance of reliable anchorage of steel bar for reinforced concrete structures. Thirty bond specimens with deformed steel bars were immersed in water and experimentally studied by pull-out tests. The soaking time from 28 day to 360 day and the cubic compressive strength of concrete with 20 N/mm^2^ and 40 N/mm^2^ were considered as the main parameters. The results indicate that the moisture content, compressive strength, and splitting tensile strength of concrete are affected by the water environment; the splitting tensile strength varies almost linearly with the compressive strength of concrete; and the descent portion of the bond–slip curve dropped slowly owing to the confinement of stirrups. On the basis of the test data, the formulas for the prediction of bond strength, residual strength, and the corresponding slips with different soaking time are proposed. Finally, the constitutive relation of bond–slip with two portions in the water environment is established with good agreement with the experimental bond–slip curves.

## 1. Introduction

Concrete is a kind of multiphase composite with microscopic pores and cracks. Owing to the wide application of concrete in civil and infrastructures, the typical working environment for concrete structures is chronically wet or exposed to water [1,2]. For example, the concrete in aqueducts, channels, piers, and coastal and offshore structures is always in a wet or saturated state as a result of the penetration of external water into the microscopic pores and cracks under the action of water pressure and liquid surface tension [2,3,4,5,6]. Studies revealed that the water not only simply penetrates into the concrete, but also generates the pore water pressure. This makes the concrete into a complex stress state affected by the humidity distribution in concrete [6,7,8]. The crack resistance and strength of the concrete decrease with the gradual increase of pore water pressure, and the wet concrete has lower static compressive strength than the dry concrete with the increase of porosity and water content in pores [9,10,11,12]. Meanwhile, at the normal curing condition with standard temperature and humidity, the compressive and tensile strengths of concrete display regular growth over time [13,14,15]. Comparatively, lower compressive strength developing of concrete results from the reduction of curing temperature and humidity, and the expected strength cannot be reached if the temperature is significantly low, regardless of curing time [16,17]. Normally, the tensile strength of wet and dry concrete increases with the loading strain rate higher than 1/s [18,19]. With the increase of moisture content in the condition of high loading strain rate over 10/s, the tensile strength of wet concrete increased [20,21]. The inconsistent reporting indicated that, although the splitting tensile strength of saturated concrete increased under rapid loading, it was lower than that of dry concrete under a static load [22]. Another investigation reported that the tensile strength of concrete decreased significantly with the increase of water content under a quasi-static load [23].

For the reliable application of reinforced concrete structures, the bond property of steel bar in concrete is a fundamental issue to be considered [2,24,25,26]. Up to now, different test methods including the pull-out test, beam-end test, and beam bond test have been used for the investigation of bond properties of steel bar in concrete. The pull-out test is commonly applied as the standard method because of the simple preparation of specimens and direct loading test [27]. The bond behavior of steel bar in concrete has been investigated considering the effects of the geometric parameters and bond length of steel bar and the enveloped conditions, and the critical length of steel bar bonded with concrete is specified in the design codes for concrete structures [2,28]. With the development of concrete technology, the bond behavior of steel bar in new concrete, such as the recycled-aggregate concrete, the manufactured sand concrete, and the lightweight-aggregate concrete, has to be studied with a comparison to that in the conventional concrete [29,30,31,32,33]. From this point of view, the changes of mechanical properties of concrete should be considered to reevaluate the bond behavior of steel bar in concrete under a water environment.

On the basis of the above review, the issue of the bond of steel bar in concrete under a water environment was put forward to address structural reliability concerns. Thirty concrete specimens with deformed steel bar were experimented using the pull-out test. The cubic compressive strength of concrete was 20 N/mm^2^ and 40 N/mm^2^, and the soaking time of specimens immersed in water varied from 28 day to 360 day. The moisture content, compressive strength, and splitting tensile strength of concrete, as well as the bond–slip curves, were measured. On the basis of the test data, the prediction formulas for the bond strength and residual strength and the corresponding slips with different soaking time are proposed, and the bond–slip constitutive model of steel bar in concrete under the water environment is suggested.

## 2. Experimental Study

### 2.1. Details of Specimen

Thirty bond specimens were designed and fabricated considering the effects of concrete strength (identified as groups A and B) and soaking time under water. According to China code GB 50152 [27], the specimens were prepared as 10 trails, and each trail had three specimens. As presented in Figure 1, the size of the specimen was 150 mm × 150 mm × 150 mm, and the bond length of steel bar was 80 mm. To avoid the influence of stress concentration at the loading end, a non-cohesive plastic tube with a length of 70 mm was set at this end. The hot-rolled HRB500 deformed steel bar with a diameter of 16 mm was centrally embedded in concrete, the yield strength was 557 N/mm^2^, and tensile strength was 724 N/mm^2^.

Two rectangular closed stirrups with a spacing of 100 mm were arranged in each specimen. In order to locate the stirrups, four constructional steel bars were tied at corners to form a frame, as shown in Figure 2. The stirrups were hot-rolled HPB300 plain steel bar with a diameter of 6 mm and the yield strength was 342 N/mm^2^. Accompanied with each group of specimens, six concrete cubes with a dimension of 150 mm were cast and cured in the same condition for the testing of compressive strength and splitting tensile strength of concrete, as per China code GB50081 [18].

The concrete was designed in two strength grades with cubic compressive strengths of 20 N/mm^2^ and 40 N/mm^2^ [2]. Ordinary silicate cements with strength grades of 32.5 and 42.5 were used as a binder for the concretes identified as groups A and B, respectively; the physical and mechanical properties are presented in Table 1. This met the requirements specified in China code GB175 [34]. The properties of fine and coarse aggregates met the requirements specified in China code JGJ 52 [35]. Natural river sand was medium-coarse with a fineness modulus of 2.91, the apparent density was 2578 kg/m^3^, and the bulk density was 1440 kg/m^3^. The crushed limestone was continuous grading with a maximum particle size of 20 mm, the apparent density was 2718 kg/m^3^, and the bulk density was 1460 kg/m^3^. Naphthalene superplasticizer was used as reducer with the measured water reduction rate of 20.4%. The dosage of raw materials was computed by the mass method for a mix proportion design of concrete [36], as presented in Table 2.

### 2.2. Specimens Soaked under Water

As displayed in Figure 3, the specimens and concrete cubes were demoulded after 1 d and soaked in the water tank at room temperature for 28 day, 90 day, 180 day, 270 day, and 360 day, respectively, before testing. The protruding surface of the steel bars was untreated during the period of water soaking because of slight corrosion.

As per China Codes GB50081 and JGJ 51 [18,37], the cube was taken out of water and moved the surface moisture, and was firstly tested for the compressive strength. After that, the sample of crushed concrete was taken as soon as possible, and the mass *m*_1_ of the concrete sample was weighed; then, the sample was put into a drying oven to be at a constant weight at a temperature of 105 °C to evaporate the free and absorbed water in the pores of the concrete, and the mass *m*_2_ of concrete sample was weighed. The moisture content *ρ* can be calculated as follows:
(1)$$\rho =\frac{m_{1}-m_{2}}{m_{2}}.$$


### 2.3. Pull-Out Test

As presented in Figure 4, the pull-out tests were carried out on a special device (Yan Gong Machinery Corporation, Zhengzhou, China) under a quasi-static load [22,23]. A load sensor was placed at the loading end to obtain the bond force of the specimen, and three displacement sensors were placed at the free end of the specimen to measure the displacements of steel bar and concrete. According to the China code SL352 [38], the loading was exerted with a rate of 7.5 kN/min until the specimen was destroyed or the slip of steel bar reached 16 mm. The load and displacement data were recorded automatically through the data acquisition system.

## 3. Results and Discussion

### 3.1. Failure Models

Two failure modes including the pull-out of steel bar and the splitting of concrete were observed in the pull-out tests. Owing to the restraint effect of stirrups on concrete, the steel bar was pulled out slowly from most of the specimens. A few specimens failed with the splitting of concrete; the splitting cracks appeared on the surface of the concrete, as exhibited in Figure 5, and the bond force dropped suddenly. With the help of stirrups, the specimens could bear a certain load before the steel bar was pulled out of concrete.

According to previous studies [24,25,26,30,31,32,33], the bond force between deformed steel bar and concrete is made up of three parts: the friction force, chemical adhesive force, and mechanical interlocking force between the transverse ribs of steel bar and concrete. Among these forces, the mechanical interlocking force accounts for the majority. As displayed in Figure 6, the bond surface of concrete was almost flat at failure owing to the effect of the transverse ribs, and the concrete powder appeared at one side of the ribs.

### 3.2. Moisture Content and Concrete Strength

Figure 7 presents the moisture content of concrete *ρ* in groups A and B, which varies with soaking time. The moisture content of concrete tends to increase with the soaking time, but the increase rate slows down in the later period. The concrete with higher strength has lower moisture content than that with lower strength.

The statistical uniform formula of moisture contents can be expressed as follows:
(2)$$\rho_{n}=11.916-0.2067f_{\mathrm{cu},28}+\left(28+2f_{\mathrm{cu},28}\right)\times 10^{-4}n-2\times 10^{-5}n^{2},$$
where $\rho_{n}$ is the moisture content at *n* days, $f_{\mathrm{cu},28}$ is the compressive strength at 28 d, and *n* is the days of soaking time (*n* ≥ 28).

On the basis of previous studies [13,14,15], the compressive and tensile strengths of concrete develop over time at the normal curing condition with standard temperature and humidity. The changes of moisture in concrete under the water environment had a certain effect on the development of concrete strength owing to the promotion of internal microcracks by penetrated water pressure. However, the development of the compressive and tensile strengths of concrete under the water environment along with the soaking time was obtained in this experiment, as presented in Figure 8. The compressive and tensile strengths of concrete under the water environment at *n* days can be predicted as follows:
(3)$$f_{\mathrm{cu},n}=f_{\mathrm{cu},28}\left[1+k_{c}\mathrm{lg}\left(n/28\right)\right],$$
(4)$$f_{\mathrm{st},n}=f_{\mathrm{st},28}\left[1+k_{t}\mathrm{lg}\left(n/28\right)\right],$$
where *f*_cu,n_ and *f*_st,n_ are the compressive and tensile strengths at *n* days under the water environment, and *k*_c_ and *k*_t_ are the statistical coefficients.

By the regression of test data, the values *k*_c_ = 0.219, *k*_t_ = 0.030 for the concrete of group A and *k*_c_ = 0.329, *k*_t_ = 0.113 for the concrete of group B were obtained. Compared with the coefficient *k*_c_ = 0.55 and 0.298 for the concrete made of cements of 32.5 and 42.5 at the normal curing condition with standard temperature and humidity [13], the growth of compressive strength was lower for group A concrete under the water environment, while that for group B concrete was similar. This is confirmed in another way in that the higher water moisture decreased the compressive strength of group A.

The relationship between the compressive strength and the splitting tensile strength can be built with an exponential function [14]. At the same soaking time, the splitting tensile strength of concrete can be predicted by the compressive strength with the following formula:
(5)$$f_{\mathrm{st},n}=af_{\mathrm{cu},n}^{b},$$
where *a* and *b* are the statistical coefficients.

By the regression of the test data of this study, *a* = 0.0684 and *b* = 1.04. Different from the coefficient *b* = 0.716 for concrete at the normal curing condition with standard temperature and humidity [14], this indicates that the splitting tensile strength of concrete under a water environment trends to be linear with the compressive strength, as illustrated in Figure 9.

### 3.3. Bond–Slip Curves

The bond–slip curves of all specimens were obtained through the central pull-out test, and the average bond stress was expressed by the following formula:
(6)$$\tau =\frac{P}{\pi \cdot d\cdot l_{a}},$$
where *τ* is the average bond stress, *P* is the pull-out force, *d* is the diameter of steel bar, and *l*_a_ is the bond length of steel bar.

The curves of three specimens in each trail were normalized to obtain the representative curve. The bond strength *τ*_u_ is taken as the average. If one of the maximum or minimum values of bond strength exceeds 15% of the average, these data are discarded and the average of the remaining two is taken as the bond strength of this trail. The test would be invalid if two values exceed 15% of the average [27]. The slip *s*_u_ corresponding to *τ*_u_ is the average slip of the valid specimens. The residual slip *s*_r_ is taken as being equal to the spacing of the adjacent cross ribs (measured for 10 mm) [31,32]. The residual strength *τ*_r_ corresponds to the residual slip *s*_r_. Typical tested bond–slip curves and the representative curve are presented in Figure 10. Similar to that of ordinary concrete confined with sitrrups, the bond–slip curve of steel bar in concrete under a water environment can be divided into the micro-slip stage, slip stage, splitting, descending stage, and residual stage. The descending stage was well measured owing to the restraint of stirrups in the specimen.

### 3.4. Bond Strength and Slip

The tested values of the bond strength *τ*_u_, residual strength *τ*_r_, and slip *s*_u_ are presented in Appendix A. As exhibited in Figure 11, the bond strength of the two groups of specimens increased with the soaking time, and reached the maximum value at 270 d. By the statistical analysis of the test results, the bond strength of steel bar in concrete under a water environment at different soaking time can be predicted as follows:
(7)$$\tau_{u,n}=\tau_{u,28}\left[1+k_{s}\mathrm{lg}\left(n/28\right)\right],$$
where *k*_s_ is 6.72 and 3.05 for groups A and B, respectively; and $\tau_{u,28}$ is the bond strength at 28 d of soaking time, which can be calculated by the following formula [24]:
(8)$$\tau_{u,28}=\left(0.82+0.9\frac{d}{l_{a}}\right)\left(1.6+0.7\frac{c}{d}+20\frac{A_{\mathrm{sv}}}{c\cdot s_{\mathrm{sv}}}\right)f_{\mathrm{st}},$$
where *c* is the thickness of concrete cover. *A*_sv_ is the cross sectional area of stirrups, *s*_sv_ is the spacing of stirrups, and *f_s_*_t_ is the splitting tensile strength of concrete.

By substitution of the test data, Formula (8) can be transformed as follows:
(9)$$\tau_{u,28}=4.7f_{\mathrm{st},28}.$$


With Formulas (7) and (9), the bond strength $\tau_{u,n}$ can be calculated. The ratio of tested to calculated bond strength is 1.078, with a variation coefficient of 0.005.

The slip *s*_u_ is a necessary parameter to determine the bond–slip constitutive model. As displayed in Figure 12, the *s*_u,n_ tends to increase with the $\tau_{u,n}^{-1}$. By regression of the test data, it can be predicted as follows:
(10)$$s_{u,n}=\frac{27.37}{\tau_{u,n}}-0.125.$$


The residual strength *τ*_r,n_ can be predicted by the bond strength with the following formula:
(11)$$\tau_{r,n}=0.20\tau_{u,n}.$$


### 3.5. Bond–Slip Constitutive Model

The constitutive model of bond–slip of the steel bar in concrete under a water environment is built by the fitting regression of the test data. The shape of the bond–slip curve is similar to that of steel bar in concrete under normal conditions, and is expressed by the following formulas:
ascent stage [25]:
(12)$$\frac{\tau }{\tau_{u}}={\left(\frac{s}{s_{u}}\right)}^{\alpha },\;0\leq s\leq s_{u},$$
descent stage [26]:
(13)$$\frac{\tau }{\tau_{u}}=\frac{s/s_{u}}{\beta {\left(s/s_{u}-1\right)}^{2}+s/s_{u}},\;s>s_{u},$$

where *α* is the coefficient of the ascent stage and *β* is the coefficient of the descent stage.

On the basis of the test data, *α* = 0.4 for the deformed steel bar. After the statistical analyses in several ways, a better correlation with R^2^ = 0.946, as displayed in Figure 13, between *β* and the ratio of *τ*_u,n_ to *s*_u,n_ is achieved:
(14)$$\beta =4.894{\left(\tau_{u,n}/s_{u,n}\right)}^{-0.724}.$$


In order to verify the adaptability of the constitutive model proposed above, the experiment bond–slip curves were compared with the predictions, as presented in Figure 14.

It can be seen that the proposed model can well evaluate the experiment curve in the ascent stage. Owing to the dispersion in the descent stage of some group specimens, the proposed curve is not so close to the tested representative curve; however, most proposed curves give a certain safety margin with lower residual bond strength.

## 4. Conclusions

On the basis of the experimental study of this paper, the following conclusions can be drawn:
(1)The moisture content increased with the soaking time owing to the penetration of environmental water into macroscope pores and cracks of concrete. A higher moisture content was reached in concrete with lower compressive strength. This induced the different changes of concrete with soaking time. Different from the exponential relationship between tensile strength and compressive strength of concrete under normal environmental conditions, the tensile strength of concrete under a water environment was linear with the compressive strength. On the basis of the test data of this study, the predictive formulas for the compressive and splitting tensile strengths of concrete under a water environment are proposed with the influencing factor of soaking time.(2)Except for a few specimens that failed with the splitting of concrete, most specimens failed with the pulling out of steel bar owing to the confining of stirrups. The bond–slip curve has a complete descending stage. As the key points of the research objectives, the bond strength, residual bond strength, and corresponding slip are provided, and the predictive formulas are built considering the soaking time.(3)The constitutive model of bond–slip with ascent and descent portions for the steel bar in concrete under a water environment is proposed, and the relevant parameters considering the soaking time are provided. The comparison with experimental bond–slip curves is exhibited to indicate the applicability of the proposed model.


## Figures and Tables

**Figure 1 materials-12-03517-f001:**
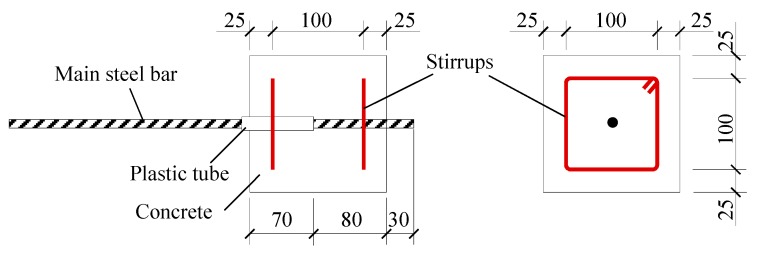
View and reinforcement details of specimens (mm).

**Figure 2 materials-12-03517-f002:**
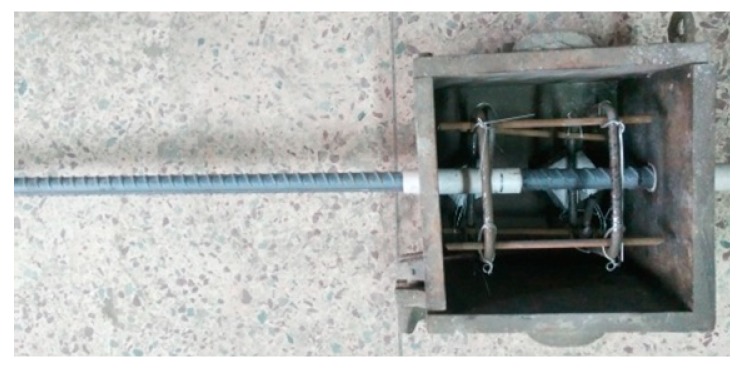
Mould for pull-out bond specimens with stirrups.

**Figure 3 materials-12-03517-f003:**
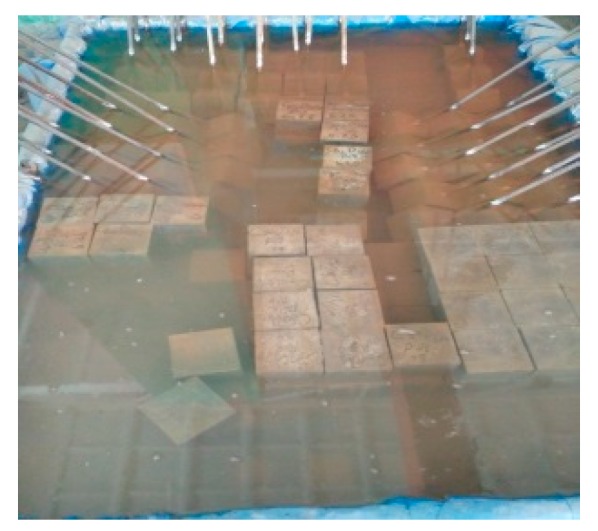
Specimens cured under water in tank.

**Figure 4 materials-12-03517-f004:**
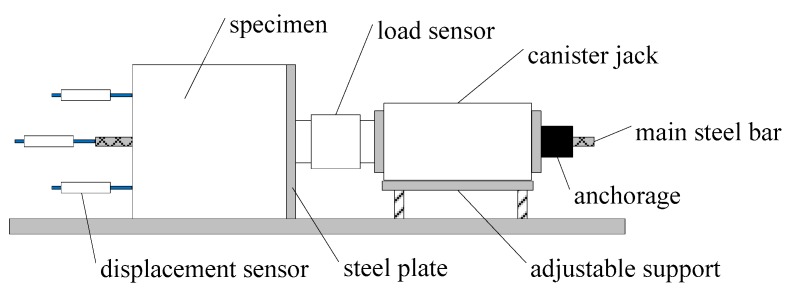
Pull-out test setup.

**Figure 5 materials-12-03517-f005:**
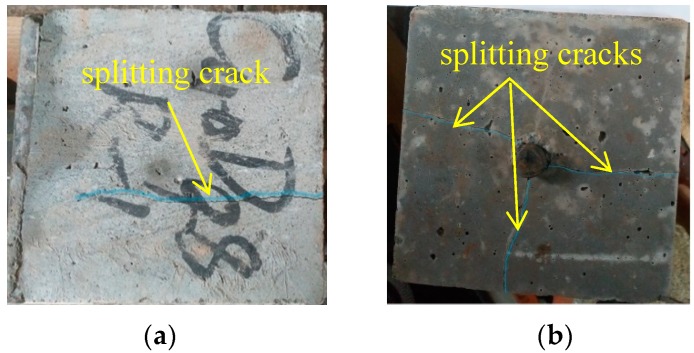
Splitting failure of concrete: (**a**) on the side surface; (**b**) on the bottom surface.

**Figure 6 materials-12-03517-f006:**
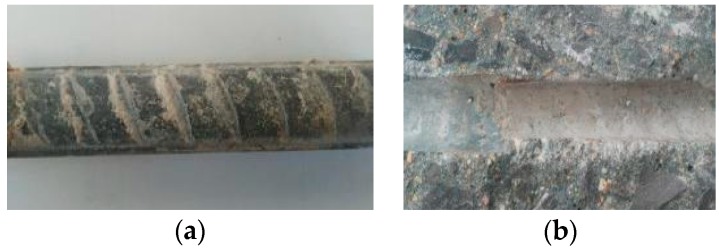
Slip surface of steel bar and concrete after failure: (**a**) steel bar surface; (**b**) concrete slip surface.

**Figure 7 materials-12-03517-f007:**
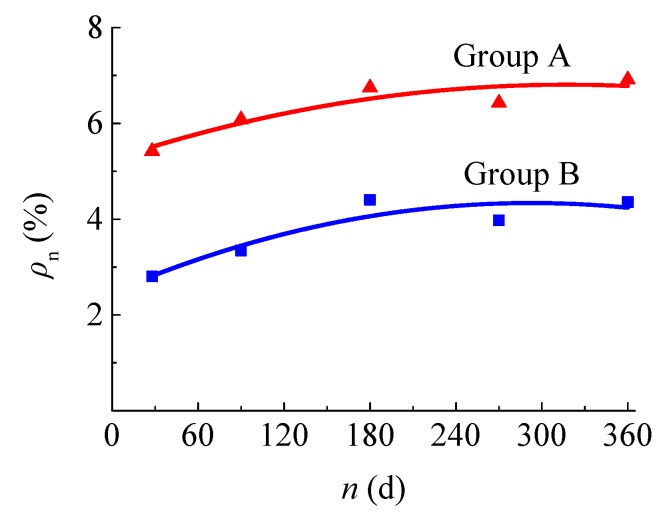
Variation of concrete moisture contents with soaking time.

**Figure 8 materials-12-03517-f008:**
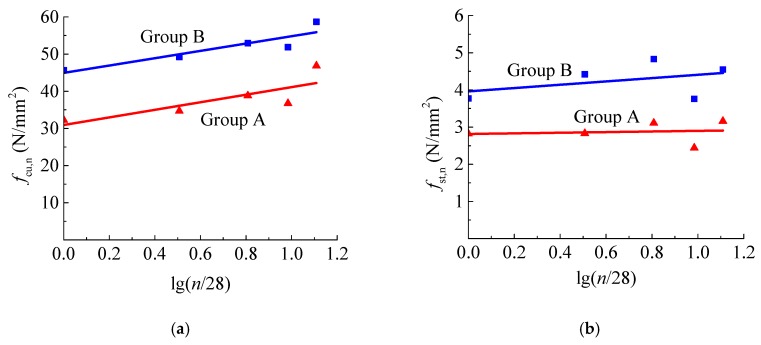
Effect of soaking time on (**a**) compressive strength; (**b**) splitting tensile strength.

**Figure 9 materials-12-03517-f009:**
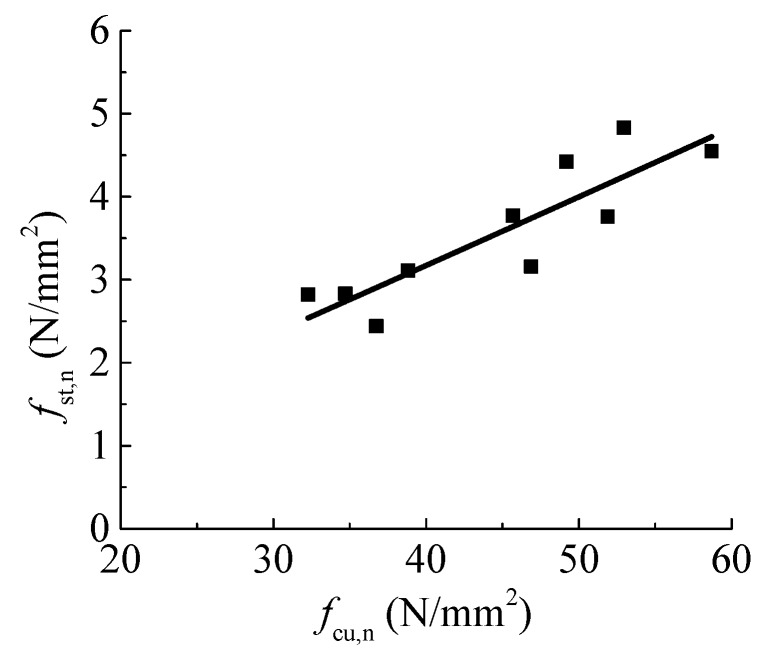
Relationship between splitting tensile and compressive strengths.

**Figure 10 materials-12-03517-f010:**
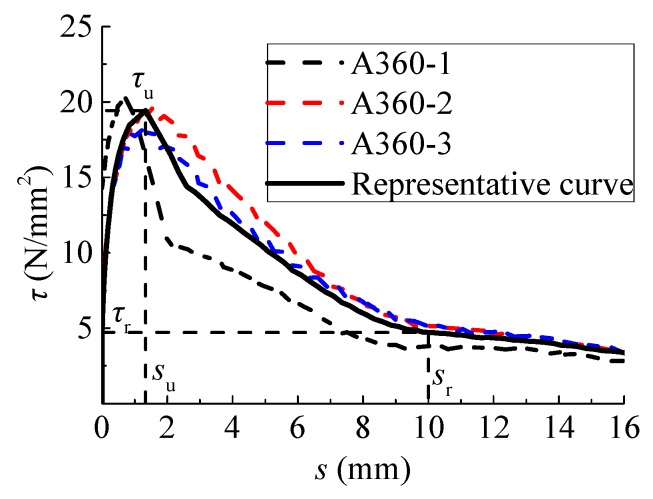
Typical experimental and representative bond–slip curves.

**Figure 11 materials-12-03517-f011:**
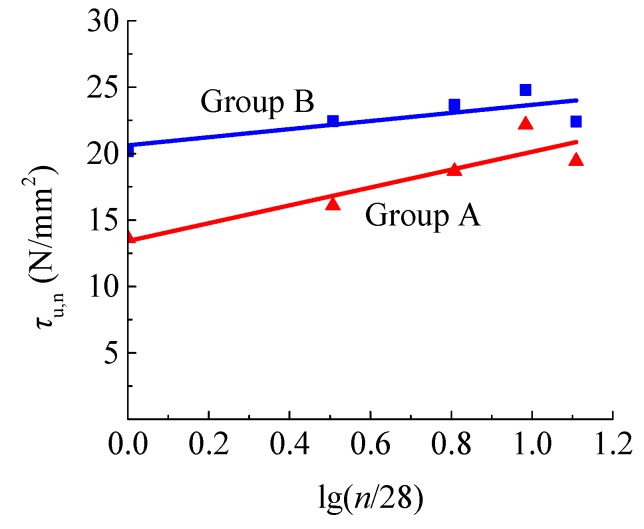
Changes of bond strength with soaking time.

**Figure 12 materials-12-03517-f012:**
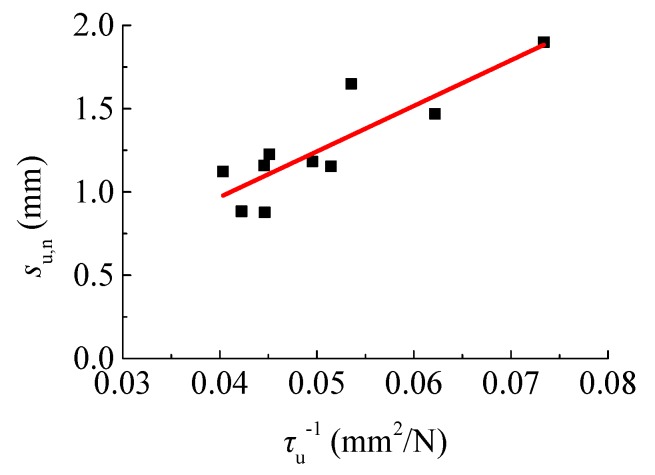
Relationship between *s*_u_ and $\tau_{u}^{-1}$.

**Figure 13 materials-12-03517-f013:**
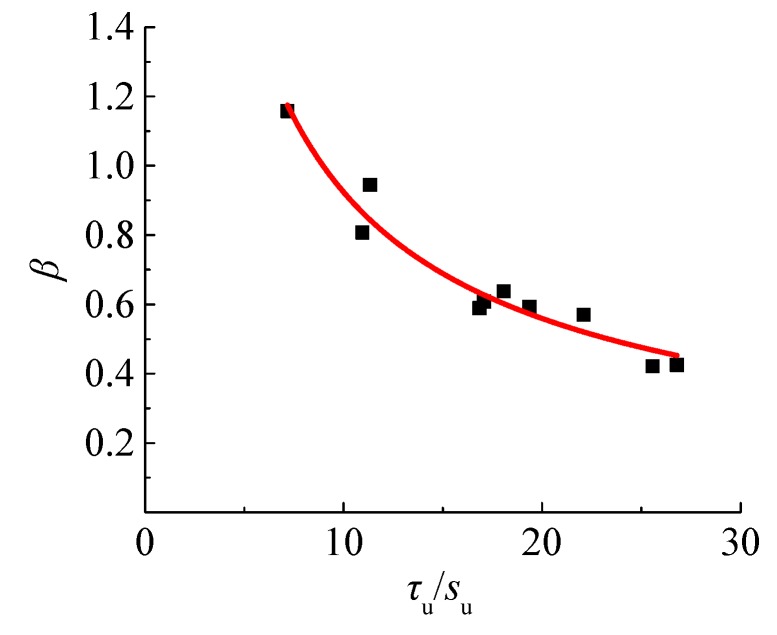
Impact of *τ*_u_/*s*_u_ on coefficient *β*.

**Figure 14 materials-12-03517-f014:**
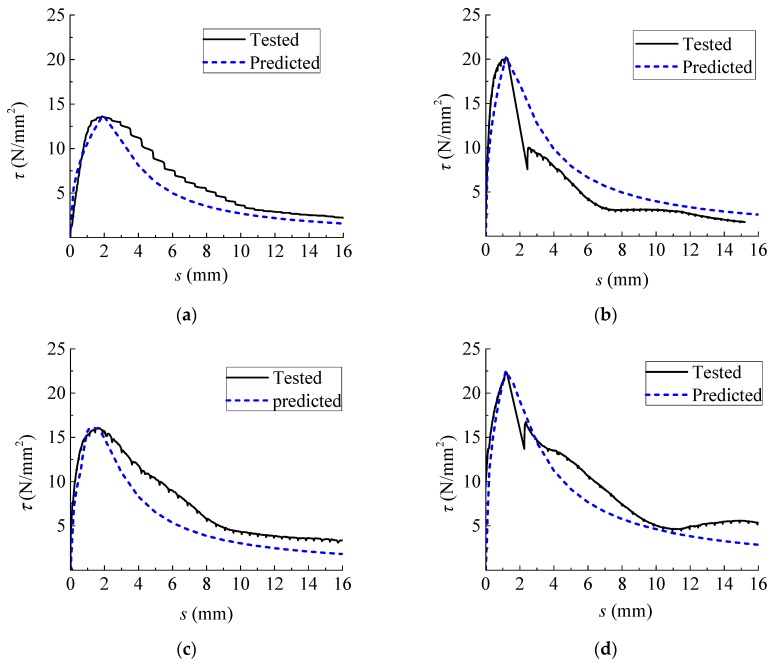
Comparisons between experimental and predicted bond–slip curves. (**a**) A28; (**b**) B28; (**c**) A90; (**d**) B90; (**e**) A180; (**f**) B180; (**g**) A270; (**h**) B270; (**i**) A360; (**j**) B360.

**Table 1 materials-12-03517-t001:** Physical and mechanical properties of cement.

Grade	Density (kg/m^3^)	Water Requirement of Standard Consistency (%)	Setting Time (min)		Compressive Strength (MPa)		Flexural Strength (MPa)	
			Initial	Final	3 Days	28 Days	3 Days	28 Days
32.5	3042	28.4	185	320	17.8	35.0	3.20	5.56
42.5	3071	26.9	165	265	28.9	45.2	4.00	6.70

**Table 2 materials-12-03517-t002:** Mixture proportion of concrete.

Group	Designed Cubic Compressive Strength (N/mm^2^)	Dosage of Raw Materials (kg/m^3^)				
		Water	Cement	Gravel	Sand	Water Reducer
A	20	210	350	1086	754	2.45
B	40	165	375	1172	688	2.63

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
