# Peer review of "Bond Properties of Steel Bar in Concrete under Water Environment"

_materials, 2019, doi:10.3390/ma12213517_

Round 1
Reviewer 1 Report
1. Describe the corrosion test method in more detail, add a photo.
2. There is a lack of information on the composition of concrete that has an effect on corrosion: the chemical composition of cement, the type and properties of the aggregate used. Only cement class (32.5 and 42.5) and max D aggregate were given. This is not enough information.
3. The designation of the concrete class C20 and C40 should be changed, such designation is unclear. Should it be C16 / 20?
4. The samples used in the study are too small. The rod cover was adopted only 2.5 cm, which is not compliant with the standard if the structure works in an environment exposed to aggression. This should be clarified.
Reviewer 2 Report
The article needs major grammatical and syntax improvements. Use of English service center is recommended. A few examples of the English errors are as follows (These are just a few examples. Even the title needs to be revised and changed. The formatting has many issues.) Bond Properties of Steel Bar with Concrete under 3 Water Environment hen strain rates was over 1.0/s [13], 42 and that of wet concrete increased with the moisture content in condition of high strain rate water environment leads a worry this aspect, rear studies were carried out. The introduction is limited at least a couple more paragraphs are needed for further discussion. It is suggested to just briefly mention about the other environmental effects and expand the introduction on the behavior of the concrete. A couple of references that is suggested for inclusion are as follow: Farzampour, Alireza. "Compressive Behavior of Concrete under Environmental Effects." IntechOpen, 2019. Farzampour, Alireza. "Temperature and humidity effects on behavior of grouts." Advances in concrete construction, An international journal 5.6 (2017): 659-669. For preparing the specimens, concrete mix design, etc. what codes are followed? This should be mentioned in each section, and the procedure and instructions for preparing the specimens should be elaborated. Image of pullout process could be helpful. The rate of loading and threshold values should be based on the recommended values in codes, if yes, the code should be mentioned within the original manuscript. Rong formatting please check the formatting rules. And, figures come after they are mentioned. How many samples for each node on the figure 5 is chosen. For the liable results at least three samples are needed for each data. The R2 value shown on the Figure 6 is very low. The regression analysis seems not to be precise. Any equations from the previous works should be referenced. More elaboration the experimental works, how did the author conduct the experiment? which specimens are used for the bond slip investigation? Figure 7 should have τ for the y axis. Figure 9 indicates an equation between Su and bond slip stress. The R2 value for regression is low; therefore, the equation could not be as practical as it is expected. Improvements on this part is suggested. Figure 11 shows the bond-slip prediction and the actual values. The predicted values are far off more than 20% in many cases and they are not either conservative nor close or accurate. It is recommended to improve the prediction equation since it could not be as practical as it is expected. Why some cases the prediction work just fine and, in some cases, they are significantly off. Generally, it is recommended to compare the results with some specimens that are not subjected to the environmental effects. Conclusion needs to be expanded with some details and quantitative results. Too many references from some authors. It is significantly recommended to reduce to at most two reference per author. Example: Li, C.Y.; Wang, F.; Deng, X.S.; Li, Y.Z.; Zhao, S.B. Testing and prediction of the strength development Li, C.Y.; Zhao, M.L.; Ren, F.C.; Liang, N,; Li, J.; Zhao, M.S. Bond Properties between full-recycled-aggregate Ding, X.X.; Li, C.Y.; Xu, Y.Y.; Li, F.L.; Zhao, S.B. Experimental study on long-term compressive strength Zhao, S.B.; Ding, X.X.; Zhao, M.S.; Li, C.Y.; Pei, S.W. Experimental study on tensile strength development Zhao, S.B.; Ding, X.X.; Li, C.M.; Li, C.Y. Experimental study of bond propertiesAuthor Response
Please see the attachment

Reviewer 3 Report
The paper presents study concerns about the bond behavior of steel bar with concrete under water environment. In my opinion, the subject of the article meets the standards of the Materials journal, however substantively there are several serious errors in the article that require accurate correction.
1. In the theoretical part, the authors rely mainly on the work of their own or the authors of their country. The results of tests on the adhesion of reinforcement to concrete in environmental conditions conducted by authors from other countries were not analyzed. Half of the cited papers are in Chinese, limiting the internationality of the literature review.
2. The description of the conditions for the formation and storage of samples in water raises great reservations, and the lack of it. This undermines the credibility of the experiment. Please describe in detail how the samples were formed and under what conditions they were stored after demoulding. Is the storage time of 28 days counted from the moment it is demoulded or after reaching C20 and C40 (i.e. after 28 days of ripening). It would be advisable to include a photo with the samples in water. Was all the reinforcement immersed in water?, Was the protruding surface of the reinforcement protected against contact with water during the maturation of the concrete. What was the water in which the samples were stored until the test?
3. What effect did the use of stirrups in the specimens on the tensile scratch? In previous works (e.g. Bond Behaviors Between Full-Recycled-Aggregate Concrete and
Deformed Steel-Bar), the authors did not use stirrups, and the nature of the destruction of the samples was similar.
4. Why two classes of plain concrete were chosen instead of, for example, plain and high-performance concrete until the test.
5. The time and methodology for drying samples raises great doubts. 12 hours is not enough to completely dry the samples. In addition, the authors did not determine the normal absorbability of the samples. The density results of saturated concrete samples were also not given.
6. The presented test results show that the presented model can be used to describe the strength of reinforcement adhesion as a function of compressive strength or shear stress. However, according to the reviewer, the authors did not achieve the basic goal, because they did not show the effect of the time of concrete maturing in water on the adhesion of reinforcement to concrete. The conclusions were not confirmed by the results of the research.
Reviewer 4 Report
Thank you for the interesting paper.
paragraph 2.1, fig 1. - how the position of the stirrups was secured without longitudinal reinforcement?
paragraph 2.2 - from the paper is not clear when soaking time have started - at the same time as concreting or 28 days after concreting (concrete cubes were soaked in the water 28 days after concreting).
lines 125-126 - you are writing that the strength of concrete in the water environment is lower than that in the air at the same age - but from formula (3) and Fig. 5 follow that it is increased, not decreased, is not it? Do I explain it wrong?
Small comment are in attached file.

Round 2
Reviewer 2 Report
the authors are required to use English service center to eliminate linguistic mistakes. thanks
Reviewer 3 Report
The authors took the reviewer's comments into account and corrected the text. The paper can be printed in Materials in the form presented.